# Action Sites and Clinical Application of HIF-1α Inhibitors

**DOI:** 10.3390/molecules27113426

**Published:** 2022-05-26

**Authors:** Renfeng Xu, Fan Wang, Hongqin Yang, Zhengchao Wang

**Affiliations:** 1Key Laboratory of Optoelectronic Science and Technology for Medicine of Ministry of Education, Provincial Key Laboratory for Developmental Biology and Neurosciences, College of Life Sciences, Fujian Normal University, Fuzhou 350007, China; xurenfeng131772@126.com (R.X.); hqyang@fjnu.edu.cn (H.Y.); 2Provincial University Key Laboratory of Sport and Health Science, School of Physical Education and Sport Sciences, Fujian Normal University, Fuzhou 350007, China; fanwang@fjnu.edu.cn

**Keywords:** HIF-1α, HIF-1α inhibitor, tumor, leukemia, disease treatment

## Abstract

Hypoxia-inducible factor-1α (HIF-1α) is widely distributed in human cells, and it can form different signaling pathways with various upstream and downstream proteins, mediate hypoxia signals, regulate cells to produce a series of compensatory responses to hypoxia, and play an important role in the physiological and pathological processes of the body, so it is a focus of biomedical research. In recent years, various types of HIF-1α inhibitors have been designed and synthesized and are expected to become a new class of drugs for the treatment of diseases such as tumors, leukemia, diabetes, and ischemic diseases. This article mainly reviews the structure and functional regulation of HIF-1α, the modes of action of HIF-1α inhibitors, and the application of HIF-1α inhibitors during the treatment of diseases.

## 1. Introduction

Hypoxia is a common phenomenon amongst physiological and pathological conditions, and is a strong stressor for cells and organisms, which can lead to metabolic disorders, and even cause cell death. In response to hypoxia stress, the body has formed a complex response mechanism. Cells adapt to hypoxia by specifically regulating the expressions of certain genes or proteins through oxygen sensors and signaling transduction pathways, in which hypoxia inducible factor-1 (HIF-1) plays an important role [1].

HIF-1 is the only transcription factor that has been found to be active under hypoxia, and is also the main nuclear transcription factor that mediates the adaptive response to hypoxia in mammals. It is closely related to the growth and development of organisms and the pathogenesis of some diseases. HIF-1 was first discovered in the nuclear extract of hypoxic hepatocellular carcinoma cell line Hep3B cells, and it can specifically bind to the hypoxia response element (HRE, 5′-RCGTG-3′) [2]. HIF-1 consists of a 120 kD HIF-1α subunit and a 91–94 kD HIF-1β subunit [3]. The β subunit, also known as the aryl hydrocarbon nuclear translocator (ARNT), is a subtype of the mammalian aryl hydrocarbon receptor complex, which is a common subunit of the HIF family (HIF1, HIF2, HIF3) [4,5,6], and its expression is constitutive and not regulated by any oxygen changes [7,8]; HIF-1α is unique to HIF-1, it is both a regulatory subunit and an active subunit of HIF-1, and its protein stability and transcriptional activity are both regulated by the intracellular oxygen concentration. Therefore, the physiological activity of HIF-1 mainly depends on the expression and activity of the HIF-1α subunit.

Studies have found that HIF-1α can control the expression of more than 100 downstream target genes, including vascular endothelial growth factor (VEGF), glucose transferase-1 (GLUT-1), erythropoietin (EPO), tyrosine kinase (c-Met), etc., involving many aspects, such as hypoxia compensatory response, immune response, angiogenesis, extracellular matrix fibrosis, transcriptional regulation, material transport, energy metabolism, etc., to increase the hypoxia tolerance of tissues. Under force, the body can better adapt to the hypoxic environment, and it can regulate many diseases (especially tumors), so HIF-1α inhibitors have potential clinical application value. In fact, based on research of the expression pattern and regulatory mechanism of HIF-1α, the design of new HIF-1α inhibitors targeting HIF-1α and its signal transduction pathway has been a hot spot of drug design in recent years, and it is also a current focus of drug design. Research hotpots include anti-tumor therapy, and the treatment of vascular disease, diabetes and other diseases. HIF-1α inhibitors mainly work by inhibiting the transcriptional activity of HIF-1α or inhibiting the upstream signaling pathway of HIF-1α. Therefore, this article mainly summarizes and discusses the mode of action of HIF-1α inhibitors and their application in disease treatment.

## 2. Structure and Function of HIF-1α Protein

HIF-1α consists of 826 amino acids, and mainly includes four functional domains (Figure 1): (1) basic helix-loop-helix domain (bHLH), which mediates the reaction with DNA element binding [9]; (2) the PAS domain, which together with bHLH mediates the binding of HIF-1β to form a heterodimer [9]; (3) two transcriptional activation domains (TAD), N-TAD and C-TAD, determined oxygen-regulated stability and transcriptional activity of the protein region; between these two TAD is the inhibitory domain (ID), which can reduce the activity of TAD of HIF-1α [10,11,12,13]; (4) oxygen-dependent degradation domain (ODDD), this domain contains N-TAD, which can control the normoxic degradation of HIF-1α through the ubiquitin–proteasome pathway [14,15,16]. In addition, HIF-1α also has two nuclear localization signals (NLS), namely NLSN (17–33 amino acids at the N-terminal) and NLSC (718–721 amino acids at the C-terminal). The HIF-1β subunit is mainly composed of three domains: bHLH, PAS, and C-TAD.

## 3. Regulation of HIF-1α Expression and Activity

HIF-1α is a subunit regulated by oxygen concentration, and there are two main oxygen-dependent pathways that regulate HIF-1α protein stability and transcriptional activity, namely, two oxygen-dependent enzymes, HIF prolyl hydroxylase (PHD) and factor-inhibiting hypoxia-inducible factor-1 (FIH-1). Three known human PHDs (PHD1, PHD2, and PHD3) can specifically hydroxylate the proline residues at positions 402 and 564 in the ODDD region of HIF-1α, and then interact with Von Hippel Lindau (VHL) protein, through binding and recruiting elongin-C/elongin-B/cullin-2 E3 ubiquitin ligase to form a ubiquitin-linked protease complex that ubiquitinates HIF-1α and degrades it via the ubiquitin-proteasome pathway [17].

FIH-1 can hydroxylate the Asp residue at the position 803 of the C-TAD region of HIF-1α, preventing HIF-1α from binding to the transcriptional coactivator histone acetyltransferase (p300)/cap binding protein (CBP), thereby inhibiting HIF-1α transcriptional activity [18]; at the same time, it can also bind to VHL and inhibit HIF-1α transactivation by recruiting histone deacetylases [19]. Therefore, under normal oxygen saturation, HIF-1α is extremely unstable and binds to VHL under the action of PHD and acetyltransferase-1 (ARD-1), resulting in HIF-1α being ubiquitinated and rapidly degraded by the proteasome. Basically, its expression is always undetectable [20,21]; however, when the oxygen concentration is lower than 5%, PHD and FIH-1 are inhibited, the ubiquitination and hydroxylation of HIF-1α decreases, the degradation of HIF-1α is hindered, and it stably exists in the cell, then combines with HIF-1β to form HIF-1, which can combine with coactivators, including CBP/p300, redox effector factor-1 (Ref-1), poly(ADP-ribose)polymerase 1 (PARP1), steroid receptor coactivator-1 (SRC-1), and TIF-2, etc., act on HRE in trans, regulate the transcription and translation of downstream genes through various post-translational modifications such as hydroxylation, acetylation, phosphorylation, ubiquitination and SUMOylation, and finally cause the reaction of corresponding pathophysiology [21,22,23].

The expression and activity of HIF-1α are not only regulated by oxygen-dependent regulation, but also co-regulated by many intracellular signaling pathways. There are two main signaling pathways, namely the phosphatidylinositol-3-kinase/protein kinase B/mammalian target of sirolimus (PI-3K/Akt/mTOR) pathway and the extracellular regulated kinase/mitogen-activated protein kinase (ERK/MAPK) pathway. The PI-3K/Akt/mTOR pathway mainly mediates the synthesis and stability of the HIF-1α protein. PI-3K is activated by certain growth factors or cytokines, such as epidermal growth factor (EGF), phosphatase, and tensin homolog deleted on chromosome 10 (PTEN), which binds to downstream Akt and phosphorylates it, thereby enhancing the translation of HIF-1α [24]. The ERK/MAPK pathway mainly mediates the function of HIF-1α transactivation. The activation of receptor tyrosine kinases such as growth factors can phosphorylate Thr796, Ser641, and Ser643 of HIF-1α through p38MAPK and ERK (p42 and p44, two kinases of MAPK signaling), and significantly upregulate the activity of HIF-1α [25,26,27]. In addition, ERK can also phosphorylate CBP/p300, thereby promoting the formation of the HIF-1α/p300 transcriptional complex and upregulating its transcriptional activation [28].

In addition to the above signaling pathways, the HIF-1-mediated hypoxia adaptive signaling pathway is also related to other signaling pathways, such as the transcription factor Myc and Notch signaling pathways related to cellular activities, and the inflammation-related NF-κB signaling pathway. Inflammatory stimulators and other factors can enhance the expression of HIF-1 gene and protein levels by regulating NF-κB-dependent signaling. In response to hypoxia, HIF-1α activates and impedes the expression of Myc target genes, leading to cell cycle arrest, and negatively regulates DNA repairs-related gene expression, inhibiting MYC transactivation by competing for binding the sites of MYC in the promoter region activated by Myc [29,30]. Studies have also found that activated HIF-1 can enhance Notch signaling by up-regulating the transcription levels of Notch ligands DLL1 and DLL4 [31]; at the same time, HIF-1α can directly act on the Notch intracellular domain (Notch ICD) to activate Notch target genes [30]; in addition, Notch ICD competes with HIF-1α for binding to FIH-1, thus affecting the FIH-1-mediated repression of HIF-1α transcriptional activity [32].

In recent years, studies have shown that some chemicals that simulate hypoxia, such as cobalt chloride (CoCl_2_), iron chelators (deferoxamine), etc., can also induce cells to express HIF-1α and produce the corresponding ischemia under normoxic conditions. Adaptation response proteins such as heat shock protein 90 (Hsp90), heat shock protein 70 (Hsp70), hypoxia-associated factor (HAF), and cyclooxygenase-2 (COX-2) can also interact with HIF-1α to mediate hypoxia signaling [33].

## 4. The Action and Mechanism of HIF-1α Inhibitors

As an important transcription factor under hypoxia, HIF-1α plays a pivotal role during growth and development, and in physiological and pathological responses in mammals. Therefore, and increasing number of people have begun to study and design HIF-1α inhibitors, and the drugs that inhibit HIF-1α activity that are currently being studied can be roughly divided into two categories (Figure 2): HIF-1α activity inhibitors and HIF-1α-related signaling inhibitors. HIF-1α activity inhibitors can be divided into the following types according to the specific mechanism of action [34,35]: (1) influence the degradation of HIF-1α; (2) inhibit the DNA transcription and expression of HIF-1α; (3) block the translation of mRNA (4) hinder the binding of HIF-1α and HRE; (5) impede the formation of HIF-1α transcription complex, etc. The present review will focus on the inhibition of HIF-1α transcriptional activity and the inhibition of HIF-1α upstream pathways.

### 4.1. Inhibit HIF-1α Transcriptional Activity

Since the discovery of HIF-1α, researchers have studied and discovered many HIF-1α inhibitors, and a variety of drugs that inhibit the transcriptional activity of HIF-1α have been used in clinical or preliminary trials (Table 1). For example, melatonin and its derivative N-butyryl-5-methoxytryptamine (NB-5-MT) promote the degradation of HIF-1α [36,37]; artepillin C and baccharin can inhibit transcriptional expression of HIF-1α [38]; MO-460 and EZN-2968 can inhibit the translational activity of mRNA [39,40]; echinomycin blocks HIF-1α binding to HRE [41,42]; and chaetocin and menadione block the formation of the HIF-1α transcription complex [43,44].

Melatonin can reduce HIF-1α protein stability by increasing PHD and VHL activity [34]. In addition, the melatonin derivative NB-5-MT can promote the interaction of HIF-1α and pVHL to promote the degradation of HIF-1α by activating the activity of PHD [37]. Manassantin A with an IC50 value of 30 nmol/L and Manassantin B with an IC50 value of 3 nmol/L extracted from Sanbaicao are relative hypoxia-specific inhibitors of HIF-1 activation, and work by blocking hypoxia-induced accumulation of nuclear HIF-1α protein to inhibit HIF-1 activity [45]. Studies have shown that curcumin with an IC50 value of 20~50 μmol/L and its derivative EF-24 with an IC50 value of 1 μmol/L promote the degradation and inactivation of HIF-1α through the proteasome pathway, and then downregulate HIF-1 downstream target genes such as EGFR and VEGF [46,47].

The four hydroxycinnamic acid analogs artepillin C, baccharin, (E)-4-(2,3-dihydrocinna-moyloxy) cinnamic acid, and drupanin extracted from Brazilian green propolis not only inhibit the transcriptional activity of HIF-1α, while inhibiting hypoxia-induced expression of target genes such as GLUT1, HK2, and VEGF, but also show no obvious cytotoxicity during the experiments [38].

Moracin O, a natural product extracted from Morus, can inhibit HIF-1α activity in Hep3B cells with an IC50 value of 6.76 nmol/L [48]. Its analog MO-460 impedes the initiation of HIF-1α translation by inhibiting binding to the 3′-untranslated region of HIF-1α mRNA [39]. The antisense oligonucleotide of HIF-1α EZN-2968 with an IC50 value of 1~5 nmol/L can specifically bind to the antagonist of RNA by locked nucleic acid technology, and it has been confirmed that it can induce the inhibition of HIF-1α mRNA and protein sustained antagonism in vitro [40]. Another factor that affects HIF-1α mRNA expression is amino flavone (AF) [34]. It is a ligand for the aryl hydrocarbon receptor (AhR), which can form dimers with HIF-1β, but inhibition of HIF-1α aggregation by AF is independent of AhR. Therefore, the mechanism by which AF inhibits HIF-1 may be achieved by regulating the expression of HIF-1α mRNA, but the exact mechanism has not been fully elucidated. The cardiac glycoside digoxin can inhibit the translation of the HIF-1α protein, but some studies have shown that it may lead to the ubiquitination and degradation of HIF-1α through the generation of ROS [49,50]. CRLX-101, originally identified as a HIF-1 inhibitor in 2002, is a nanoparticle consisting of water soluble cyclodextrin based polymers containing pendant carboxylic acid groups, camptothecin (CPT), and alternating repeating polyethylene glycol (PEG) blocks, and it inhibits the accumulation of HIF proteins [51]. EZN-2208 is a water-soluble polyethylene glycol conjugate of SN38 (10-hydroxy-7-ethyl-camptothecin), an analogue of topoisomerase I inhibitor CPT-11, which has the effect of inhibiting HIF-1α, and further down-regulating the target genes of HIF-1α, especially VEGF [52].

The estrogen metabolite 2-methoxyestradiol (2-ME2) has been widely studied as an anticancer agent, it can inhibit growth and induce apoptosis of osteosarcoma, esophageal cancer, ovarian cancer, nasopharyngeal cancer, prostate cancer and other tumors in vitro and in vivo [53]. Studies have found that it can down-regulate the HIF-1α protein level and HIF transcriptional activity in an oxygen and proteasome-independent manner and interfere with the expression of the HIF-1α downstream target gene VEGF [54].

Echinomycin (NC-13502) of the quinoxaline family works by inhibiting the binding of HIF-1α to the HRE in the promoter region of VEGF, and can also inhibit the binding of HIF-1α to DNA [41,42].

The natural product piperazinone compound chaetocin with an IC50 value of 12.5 μmol/L extracted from the fungus of the genus Nigella mainly inhibits HIF-1 activity by hindering HIF-1α and HIF-1β two subunits, and polymerizing and inhibiting the interaction of HIF-1α with p300 [40]. The study also found that menadione with an IC50 value of 40.6 ± 3.2 μmol/L and ethacrynic acid with an IC50 value of 234.4 ± 6.5 μmol/L have the effect of blocking HIF-1α/p300 PPI [55]. The benzopyran compound 103D5R blocks the binding of HIF-1α/ARNT to target genes by competitively binding to p300, thereby blocking the expression of target genes [44].

### 4.2. Inhibit HIF-1α Upstream Pathways

Acting on each link in a HIF-1α-related signaling pathway can also inhibit the activity of HIF-1α, such as the common PI-3K/Akt/mTOR inhibitors such as rapamycin and its analog everolimus; the common MAPK inhibitors such as PD98059 Sorafenib (BAY43-9006), cyclooxygenase 2 inhibitor and celecoxib; the common receptor tyrosine kinase inhibitors trastuzumab, VEGF receptor 2 (VEGFR2) tyrosine kinase inhibitor, and epidermal growth factor receptor tyrosine kinase inhibitor, etc. In recent years, it has been reported that microtubule-targeting agents and topoisomerase inhibitors such as topotecan and camptothecin can inhibit the stability and transcriptional activity of HIF-1α [54].

The PI-3K inhibitors wortmanin and LY294002 bind to PI-3K, inhibit the PI-3K/Akt signaling pathway and the translation of HIF-1α. VEGFR2 tyrosine kinase inhibitors such as AAL993 and AG1478 can inhibit HIF-1α transcriptional activity by blocking Akt and/or ERK phosphorylation signaling [56,57]. Benzopyrazoles are a new class of HIF-1α inhibitors, which have two mechanisms of action. One is to block HIF-1 and p300/alpha by stimulating FIH to hydroxylate the C-TAD terminal of HIF-1α and the binding of CBP, so that the target genes cannot be expressed. The other is to regulate the expression of HIF-1α at the level of mRNA translation by inhibiting PI-3K/Akt/mTOR/4E-BP signaling, without affecting the expression of HIF-1α at the cellular level [58]. 3-(5′-Hydroxymethyl-2’-furyl)-1-benzylindazole (YC-1) with an IC50 value of 1.2 μmol/L, which also belongs to this class of compounds, was discovered and identified in 2001 as a HIF-1α-specific inhibitor for the first time. It does not alter the degradation of HIF-1α mRNA level, but it can regulate HIF-1α expression at the translational level by inhibiting the PI-3K/Akt/mTOR/4E-BP pathway [59,60,61].

A specific inhibitor of HSP90, 17-allylaminogeldanamycin (17-AAG) can inhibit its activity by competitively inhibiting the binding of HSP90 and HIF-1α, activating the ubiquitination degradation of HIF-1α, destroying the stability of HIF-1α, and finally inhibiting the expression of downstream target gene VEGF [62,63]. The aryloxy acetamido-based structural analog AC1-004 with an IC50 value of 1.8 μmol/L destabilizes HIF-1α and degrades it by inhibiting the action with HSP90. In an antitumor test in mice, AC1-004 significantly reduced the tumor size by about 58.6% without adverse reactions such as weight loss, and is expected to become a new structural type of HIF-1α inhibitor [64].

**Table 1 molecules-27-03426-t001:** Action Sites of HIF-1α Inhibitors.

HIF-1α Inhibitor	Chemical Formula	Action Site	Reference
Melatonin	C_13_H_16_N_2_O_2_	Promote the degradation of HIF-1α	[34]
NB-5-MT	C_15_H_20_N_2_O_2_	Promote the degradation of HIF-1α	[35]
Manassantin A	C_42_H_52_O_11_	Decrease HIF-1α protein accumulation	[43]
Manassantin B	C_41_H_48_O_11_	Decrease HIF-1α protein accumulation	[43]
EF-24	C_19_H_15_F_2_NO	Promote the degradation of HIF-1α	[44]
Curcumin	C_21_H_20_O_6_	Promote the degradation of HIF-1α	[45]
Artepillin C	C_19_H_24_O_3_	Inhibit HIF-1α transcriptional activity	[31]
Baccharin	C_29_H_38_O_11_	Inhibit HIF-1α transcriptional activity	[31]
Moracin O	C_19_H_18_O_5_	Inhibit the translation activity of HIF-1α mRNA	[46]
MO-460	C_19_H_18_O_4_	Inhibit the translation activity of HIF-1α mRNA	[37]
AF	C_15_H_11_NO_2_	Inhibit the translation activity of HIF-1α mRNA	[32]
Digoxin	C_41_H_64_O_14_	Inhibit the translation of HIF-1α; promote the degradation of HIF-1α	[47,48]
EZN-2208	C_104_H_111_N_12_O_37_	Inhibit the translation activity of HIF-1α mRNA	[50]
2-ME2	C_19_H_26_O_3_	Inhibit HIF-1α transcriptional activity; decrease HIF-1α protein accumulation	[51,52]
Echinomycin	C_51_H_64_N_12__0__12_S_2_	Hinder HIF-1α binding with HRE	[39,40]
Chaetocin	C_30_H_28_N_6__0__6_S_4_	Impede the formation of HIF-1α transcription complex	[38]
Menadione	C_11_H_8_O_2_	Impede the formation of HIF-1α transcription complex	[53]
Ethacrynic acid	C_13_H_12_C_l__2__0__4_	Impede the formation of HIF-1α transcription complex	[53]
103D5R	C_20_H_21_N_3_O_2_	Impede the formation of HIF-1α transcription complex	[42]
AAL993	C_20_H_16_F_3_N_3_O	Inhibit Akt and/or ERK signaling pathway	[54]
AG1478	C_16_H_14_ClN_3__0__2_	Inhibit Akt and/or ERK signaling pathway	[55]
YC-1	C_19_H_16_N_2_O_2_	Inhibit the PI-3K/Akt/mTOR/4E-BP pathway	[57,58,59]
17-AAG	C_31_H_43_N_3_O_8_	Inhibit HIF-1α binding with HSP90	[60,61]
AC1-004	C_25_H_27_N_2_O_3_	Inhibit HIF-1α binding with HSP90	[62]

## 5. Clinical Application of HIF-1α Inhibitors

According to present research, HIF-1α has low expression in the human brain, lung, placenta, heart, skeletal muscle, kidney, and pancreas under normoxia, but there is an exponentially large increase of HIF-1α in the brain, lung, kidney, heart, and other tissues under hypoxia [65]. HIF-1α inhibitors can be widely used in the treatment of various diseases related to HIF-1 overexpression, such as tumors; leukemia; diabetes and its complications; ischemic, cardiovascular and brain diseases; and inflammatory diseases, etc.

### 5.1. Anti-Tumor Therapy

The relationship between HIF-1α and tumors is a hot research topic at present. It has been found that HIF-1α is overexpressed in a variety of human cancers, including bladder, liver, breast, lung, osteosarcoma, glioma, ovarian, prostate, colon, and renal cell carcinoma [66,67,68]. During the occurrence and development of tumors, the growth speed of malignant tumor cells is uncontrolled and too fast, while the rate of tumor angiogenesis is relatively lagging, finally resulting in insufficient blood supply and widespread hypoxia in the tumor. HIF-1α is the main hypoxia response factor, so this anoxic environment leads to the increase of HIF-1α expression [69]. It has been reported that the expression level of HIF-1α in tumor specimens is positively correlated with the aggressiveness and poor prognosis for cancer patients with conventional therapy [70]. HIF-1 activates the transcription of genes involved in key aspects of cancer biology, and regulates the expression of many downstream genes, including hypoxic energy metabolism (PDK1 and LDHA), angiogenesis (VEGF and EPO), intracellular matrix remodeling (MMP1 and LOX), apoptosis/autophagy (BNIP3 and NIX), cell survival (Myc and IGF family) and cell invasion/migration and escape (CXCR4), etc. All of these make tumor cells resistant to hypoxia, and increase oxygen/energy supply, thereby promoting tumor growth, invasion, and metastasis [71,72]. In preclinical studies, inhibition of HIF-1α activity by various methods and down-regulation of various HIF-1α-mediated gene expressions have shown a significant impact on tumor growth. Therefore, the use of specific small-molecule inhibitors targeting HIF-1α is an attractive strategy for developing cancer therapeutics [73]. Efforts are currently underway to identify HIF-1α specific inhibitors and examine their efficacy as anticancer therapeutics.

Preclinical studies have confirmed that echinomycin has antitumor activity in liver cancer, breast cancer, colon cancer, and other diseases. Manassantin A and Manassantin B have strong selective inhibitory effects on human breast cancer T47D cells, and also have strong inhibitory effects on secretion of hypoxia-induced VEGF, CDKN1A, and GLUT-1 genes [45,74]. Studies have shown that curcumin and its derivative EF-24 have the effect of inhibiting tumor blood-vessel growth and delaying tumor growth [46,47]. YC-1 is a targeted HIF-1α inhibitor, which can resist intravascular thrombosis, block the hypoxia signaling pathway of cells, inhibit the expressions of HIF-1α and VEGF, inhibit tumor angiogenesis, and tumor cell proliferation, thereby exerting anti-tumor and anti-angiogenic effects [75,76]. Therefore, YC-1 can also be used to treat various diseases related to the overexpression of HIF-1α, such as angiogenesis-related diseases, immune disorders, cardiovascular remodeling, and pulmonary hypertension [58,77,78]. Sun et al. confirmed that intratumoral HIF-1α expression was inhibited by the transfection of antisense HIF-1α gene plasmid, resulting in the downregulation of VEGF and the reduction of intratumoral vascular density [79]. In addition, two applicable HIF-1α cDNA variants were recently discovered, both of which have bHLH and PAS domains, but lack the ODDD and TAD domains, one is HIF-1αZ, which is induced by zinc and competitively dimerizes with HIF-1β to inhibit the activity of HIF-1α, but does not block the nuclear translocation of endogenous HIF-1α [80]; the other is HIF-1α alternative splicing variant HIF- 1α516, a translated polypeptide of 516 amino acid residues, which also inhibits HIF-1α activity by competing with endogenous HIF-1α and binding to HIF-1β [81].

### 5.2. Leukemia Therapy

Various types of leukemia have an abnormally high expression of the HIF-1α gene, so the study of HIF-1α inhibitors may become a new strategy for the treatment of leukemia. In chronic lymphocytic leukemia (CLL), HIF-1α promotes bone marrow neovascularization, regulates the expression of chemokine receptors and cell adhesion molecules (such as CXCR 4, CXCL 12, etc.), thereby maintaining CLL cells survival within the bone marrow microenvironment [82]. Abdul-Aziz et al. [83] found that silencing HIF-1α inhibited the transcriptional regulation of metastasis suppressor factor (MIF) in AML cells in bone marrow under hypoxic conditions, thereby improving the survival of AML cells, indicating the pivotal role of the hypoxia/HIF-1α/MIF axis in promoting the survival and proliferation of an AML tumor. Zhang et al. [84] found that deficiency of HIF-1α in chronic myeloid leukemia (CML) resulted in increased expression of the cell cycle inhibitors (p16InK4a, p19Arf, and p57) and the apoptosis gene (p53) in leukemia stem cells (LSC). Ng et al. [85] demonstrated that the maintenance of CML stem cells under hypoxic conditions is dependent on HIF-1α by silencing HIF-1α gene. In patients with T-cell acute lymphoblastic leukemia (T-ALL), HIF-1α is overexpressed, and HIF-1α can promote the activation of Notch1 signaling, increase the expression of cyclin D1, CDK2, p21, MMP2, and MMP9 proteins, thereby leading to cell proliferation, invasion, and resistance [86,87]. Few data are available on the role of HIF-1α in acute B-lymphoblastic leukemia, but it has been demonstrated that HIF-1α expression is induced by leukemic B cells in the bone marrow [88].

Valsecchi et al. treated CLL with EZN-2208 and found that EZN-2208 can significantly inhibit the expression of HIF-1α, thereby impairing the directed chemotaxis of CXCL 12 in CLL cells, and induce cell death and prolong the survival time of mice with significant anti-leukemia activity [82,89,90]. In addition to being an anticancer agent, 2-ME2 also has anti-leukemia activity. The possible mechanisms of action include inhibiting the accumulation of superoxide dismutase (SOD) and reactive oxygen species (ROS), increasing the expressions of p53 and p21, and promoting the growth of cells in G2/M-phase cell cycle arrest, etc., thereby inhibiting the levels of HIF-1α, microtubules and tumor angiogenesis [91]. In all, 2-ME2 can inhibit the translation of MYC, a downstream effector of Notch1, and prevent the activity of SCL/TAL1, thereby eliminating the self-renewal activity of pre-leukemia stem cells. It can also block the G2/M-phase cell cycle, causing acute T lymphocytes to show typical apoptotic changes [53,91]. Similarly, in AML, 2-ME2 can inhibit the expressions of HIF-1α, VEGF, GLUT1, and HO-1.

Although the research on HIF-1α inhibitors in leukemia is at an early stage, based on the deepening research and the functional analysis of hypoxia-inducible factor in leukemia in the existing literature, HIF-1α inhibitors are expected to be a reliable treatment option for leukemia patients.

### 5.3. Diabetes and Its Complications

In diabetic tissues, hypoxia triggered by insufficient activation of HIF-1α signaling and impaired adaptive responses to hypoxia are fundamental pathogenic factors during the development of diabetes and diabetic complications [92]. Hyperglycemia-induced methylglyoxal modifies p300 and disrupts the binding of p300 to HIF-1α, thereby leading to the destabilization of HIF-1α and the downregulation of HIF-1α-related responses under hypoxic conditions [93]. Paradoxically, hyperglycemia also activates HIF-1α signaling in glomerular mesangial cells, suggesting that HIF-1α regulation is context-specific in diabetes [94]. In addition, HIF-1α is also associated with the development of microvascular and macrovascular complications in diabetes [95]. Therefore, strategies aimed at modulating HIF-1α signaling may be promising new treatments for diabetes and its complications.

Studies have shown that the use of deferoxamine (DFO) can eliminate methylglyoxal binding, correct high glucose-induced impairment of HIF-1α/p300 binding, and normalize the transactivation of HIF-1α [93]. According to the results of numerous studies, curcumin has also been used to treat diabetes and alleviate its complications by targeting multiple pathways [96].

### 5.4. Ischemic Cardiovascular and Cerebral Disease

In recent years, both basic and clinical studies have shown that myocardial ischemia and cerebral ischemia can significantly increase the expressions of HIF-1α mRNA and protein, which are undetectable in normal ventricular myocardium. Experimental evidence has shown that HIF-1α plays a key role in the protective effect of ischemic preconditioning in both myocardial and cerebral ischemic injury. In a femoral artery ligation model with HIF-1α gene knockout, it was found that vascular growth factors such as VEGF were not activated, and the ability of reperfusion after ischemia was reduced [97]. The proangiogenic effect of HIF-1α has also been found in injury models such as myocardial hypertrophy, myocardial infarction, and wound healing [98,99]. Therefore, in view of the important role of HIF-1α in various ischemic diseases, it may provide a new and effective therapeutic approach for diseases characterized by hypoxia, which has attracted extensive attention of scholars. Studies have found that S-nitrosoglutathione (GSNO) treatment stabilizes HIF-1α and induces the downstream gene expression of HIF-1α targets to stimulate regenerative processes, resulting in functional recovery in animals with mild traumatic brain injury [100].

### 5.5. Other Disorders

The expression of HIF-1α can be detected in inflammatory diseases such as immune inflammation, bacterial infection, macrophage metabolism and viral infection, as well as in corresponding inflammatory sites in patients with arthritis, arteriosclerosis, and autoimmune diseases. When inflammation occurs, local vascular permeability is enhanced, causing immune cells to aggregate at the site of inflammation, in a rapidly hypoxic environment, which in turn induces immune cells to transcribe HIF-1α. HIF-1α plays a key role in the synthesis of pro-inflammatory factor interleukin-1β (IL-1β) [101]. In macrophages and neutrophils, HIF-1α can activate NF-κB [102]. A compound extracted from celery, 3-N-Butylphthalide (NBP), may have anti-inflammatory effects by inhibiting the formation of HIF-1α transcriptional complex [103].

Many studies have also found that HIF-1α plays a certain role in the pathogenesis of rheumatoid arthritis (RA). Hu et al. found that HIF-1α makes the relationship between RA synovial fibroblasts (RASFs) and T/B cells persistent [104]. HIF-1α interacts to induce the production of inflammatory cytokines and autoantibodies, thereby aggravating the development of RA [102]. Similarly, Hu et al. also found that HIF-1α can enhance the expression of IL-8, IL-33, MMP, and VEGF in RASFs [105], which aggravates inflammation, cartilage destruction, and angiogenesis, and participates in the pathogenesis of RA.

HIF-1α is also involved in the pathogenesis of systemic lupus erythematosus (SLE). HIF-1α can induce SLE by affecting the ratio of Th17 and Treg cells, thereby causing an immune imbalance between these two cells [106]. Induced proliferation-related signaling protein 1 can cause pathological changes in the kidneys of SLE patients by regulating the expression of HIF-1α [107]. MiRNA-210 regulates the expression of HIF-1α and the differentiation of Th17 cells when the body is under hypoxic condition, destroying the body function of SLE patients [108]. These findings suggest that HIF-1α may be a new diagnostic marker for SLE. Although there are few applications of HIF-1α inhibitors in the treatment of SLE at present, in-depth research on HIF-1α inhibitors will lay an evidence-based medical foundation for the development of HIF-1α-targeted SLE treatment.

## 6. Conclusions

Together, as an important transcription factor under hypoxia, HIF-1α regulates angiogenesis, glucose metabolism, apoptosis, and autophagy, and also participates in the regulation of multiple signaling pathways, playing an important role during the growth and development of the body, and in various physiological and pathological processes. Therefore, the regulation of HIF-1α activity may be a breakthrough point for the treatment of many diseases. Up-regulation of HIF-1α activity can improve cell viability under hypoxic conditions and increase angiogenesis in hypoxic tissues. On the contrary, HIF-1α inhibitors can prevent angiogenesis and reduce the viability of hypoxic or inflammatory tissues. Currently, most hypoxia-inducible factor inhibitors have been studied in solid tumors, with a small number being tested in preclinical models of hematological malignancies. We believe that with the in-depth research on the transcriptional mechanism of HIF-1α and its mediating genes, the therapeutic direction for HIF-1α will receive more and more attention, and there will be many good HIF-1α inhibitors in the near future for clinical research, creating a new direction for disease prevention and treatment.

## Figures and Tables

**Figure 1 molecules-27-03426-f001:**
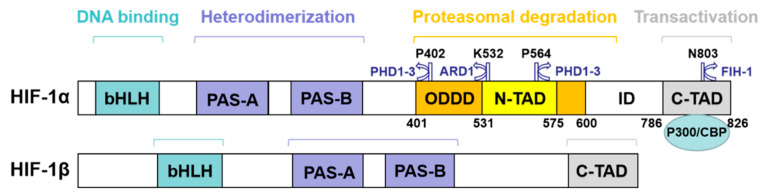
Schematic representation of HIF-1 structure and function.

**Figure 2 molecules-27-03426-f002:**
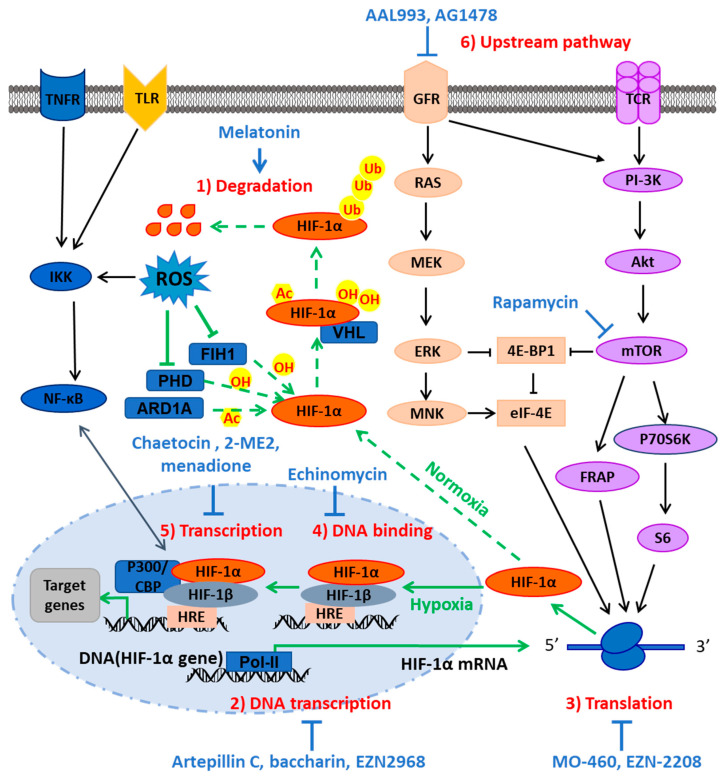
The signaling pathways of HIF-1α inhibitors and their actions. HIF-1α inhibitors mainly inhibit HIF-1α activity through the following mechanisms: (1) influence the degradation of HIF-1α; (2) inhibit the DNA transcription and expression of HIF-1α; (3) block the translation activity of mRNA; (4) hinder HIF-1α binding with HRE; (5) impede the formation of HIF-1α transcription complex; and (6) inhibit HIF-1α upstream pathway. Dashed arrows indicate the response under normoxia, solid arrows indicate the response under hypoxia, green arrows indicate the regulation mechanism of HIF-1a self-activity, and black arrows indicate the regulation mechanism of signaling pathway.

## Data Availability

Not applicable.

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
