# Peer review of "Action Sites and Clinical Application of HIF-1α Inhibitors"

_molecules, 2022, doi:10.3390/molecules27113426_

Round 1
Reviewer 1 Report
This manuscript summarized action mechanisms of some HIF-1a inhibitors. They also described application of anti-HIF-1a drugs to some human diseases. It may be useful for readers who wish to look over how HIF-1a inhibitors function and how HIF-1a inhibition can be clinically applicable. However, this reviewer has many concerns as listed bellow.
- Authors stated “The present review will focus on the inhibition of HIF-1a transcriptional activity and the inhibition of HIF-1a upstream pathways” (page 3, line 139-140). However, so far, many review papers regarding HIF inhibitors and its application have been published. Authors should clearly state what is new in this manuscript.
- Introduction: page 1, line 38: “which is a common subunit of several transcription factors” This needs references.
- A sentence indicated in comment #1 (line 139-140. Page 3) should be moved to Introduction section.
- Page 3, line 102: Authors stated “translational apoptosis”. I did not understand it. Additionally, no apoptosis data provided in ref 21.
- Figure 2: Authors clarified mechanisms of HIF-1a inhibitors in the text and figure legend. However, the given numbering was not reflected in the illustration.
- Figure 2: Illustration demonstrated that rapamycin inhibits PI-3K. However, it is an inhibitor of mTORC complexes.
- Figure 2: Authors need to explain what is difference between solid, dashed, green, and black arrows.
- Page 5, line 192: Rapamycin is identical to sirolimus. Everolims is a derivative of rapamycin.
- Page 5, lines 202-207: A reference provided (47) is inappropriate for benzopyrazole story. New appropriate reference(s) are required.
- Page 5, line 210: Authors stated that YC-1 is HIF-1a specific. Is this true? YC-1 is known to has many mechanisms of action including HIF-1a translational inhibition.
- Page 6, lines 222-225: Ref 54 seems inappropriate for these statements.
- Page 6, lines 253-257: Manassantins and curcumin was not included in section 4.
- Page 6-7, lines 264-266: Statement regarding clinical application of YC-1 needs references.
- Page 7 lines 282-286: References missing regarding AML story.
- Page 7, lines 297-301: authors discussed about EZN-2208 and 2-ME2. However, these drugs were not included in Figure 1 and section 4.
- Section 5.3: DFO (deferoxamine) and desferrioxamine (page 3, line 122) are identical and need to be unified in this manuscript.
Author Response
Authors stated “The present review will focus on the inhibition of HIF-1a transcriptional activity and the inhibition of HIF-1a upstream pathways” (page 3, line 139-140). However, so far, many review papers regarding HIF inhibitors and its application have been published. Authors should clearly state what is new in this manuscript.
Response: This review summarizes the action mechanism of HIF-1a inhibitors, describes the application of anti-HIF-1a drugs in some human diseases, and lists some HIF-1a inhibitors that have been used in the clinic and newly discovered HIF-1a inhibitors, which will help to further understand how HIF-1a inhibitors work and how they are used clinically, and grasp the latest developments of HIF-1a inhibitors. Thank you very much for your valuable comments.
Introduction: page 1, line 38: “which is a common subunit of several transcription factors” This needs references.A sentence indicated in comment #1 (line 139-140. Page 3) should be moved to Introduction section.
Response: The Reference has been added (Chen, Y.; Gaber, T. Hypoxia/HIF modulates immune responses. Biomedicines. 2021, 9(3), 260.). Thank you very much.
Page 3, line 102: Authors stated “translational apoptosis”. I did not understand it. Additionally, no apoptosis data provided in ref 21.
Response: The literature has been consulted, and confirmed that it is enhanced translation, no apoptosis, which has been modified in the revised manuscript. Thank you very much.
Figure 2: Authors clarified mechanisms of HIF-1a inhibitors in the text and figure legend. However, the given numbering was not reflected in the illustration.
Response: The corresponding numbers have been added in Figure 2 of our revised manuscript. Thank you very much.
Figure 2: Illustration demonstrated that rapamycin inhibits PI-3K. However, it is an inhibitor of mTORC complexes.
Response: Already edited in our revised manuscript. Thank you very much.
Figure 2: Authors need to explain what is difference between solid, dashed, green, and black arrows.
Response: Dashed arrows indicate the response under normoxia, solid arrows indicate the response under hypoxia, green arrows indicate the regulation mechanism of HIF-1a self-activity, and black arrows indicate the regulation mechanism of signaling pathway. Thank you very much.
Page 5, line 192: Rapamycin is identical to sirolimus. Everolims is a derivative of rapamycin.
Response: Relevant content has been modified in our revised manuscript. Thank you very much.
Page 5, lines 202-207: A reference provided (47) is inappropriate for benzopyrazole story. New appropriate reference(s) are required.
Response: New appropriate references have been added in our revised manuscript. Thank you very much.
Page 5, line 210: Authors stated that YC-1 is HIF-1a specific. Is this true? YC-1 is known to has many mechanisms of action including HIF-1a translational inhibition.·
Response: Yes, we have confirmed that YC1 is relatively specific to HIF-1a. It is a molecule with various physiological functions, and its targets are VEGF and NF-κB in addition to HIF-1a. Thank you very much.
Page 6, lines 222-225: Ref 54 seems inappropriate for these statements.
Response: The statements have been revised and the new appropriate references have also been added in our revised manuscript. Thank you very much.
Page 6, lines 253-257: Manassantins and curcumin was not included in section 4.·
Response: Manassantins and curcumin related content has been added to Section IV in our revised manuscript. Thank you very much.
Page 6-7, lines 264-266: Statement regarding clinical application of YC-1 needs references.·
Response: References have been added in our revised manuscript. Thank you very much.
Page 7 lines 282-286: References missing regarding AML story.·
Response: References have been added in our revised manuscript. Thank you very much.
Page 7, lines 297-301: authors discussed about EZN-2208 and 2-ME2. However, these drugs were not included in Figure 2 and section 4.·
Response: EZN-2208 and 2-ME2 have been added into Figure 2 and Section IV in our revised manuscript. Thank you very much.
Section 5.3: DFO (deferoxamine) and desferrioxamine (page 3, line 122) are identical and need to be unified in this manuscript.·
Response: Unified in our revised manuscript. Thank you very much.

Reviewer 2 Report
In this review article by Zhengchao et al. authors discussed action sites and clinical applications of HIF-1a inhibitors. The authors discussed the HIF-1a pathway and its possible roles & applications. Although it’s a review on HIF-1a inhibitors which is one of the hot topics in research, I found it not informative enough than literature already present. As a review article authors need to add more information than the general information already discussed in previously published reviews.
For example, the below review article which the authors failed to add has the most detailed review published in 2015.
Masoud GN, Li W. HIF-1α pathway: role, regulation, and intervention for cancer therapy. Acta Pharmaceutica Sinica B. 2015 Sep 1;5(5):378-89.
Should have some information about inhibitors for these discussed applications.
Author Response
In this review article by Zhengchao et al. authors discussed action sites and clinical applications of HIF-1a inhibitors. The authors discussed the HIF-1a pathway and its possible roles & applications. Although it’s a review on HIF-1a inhibitors which is one of the hot topics in research, I found it not informative enough than literature already present. As a review article authors need to add more information than the general information already discussed in previously published reviews. For example, the below review article which the authors failed to add has the most detailed review published in 2015. Masoud GN, Li W. HIF-1α pathway: role, regulation, and intervention for cancer therapy. Acta Pharmaceutica Sinica B. 2015 Sep 1;5(5):378-89. Should have some information about inhibitors for these discussed applications.
Response: We have added some newer information from recent references in our revised manuscript and supplemented some information discussed in the application. Thank you very much for your valuable comments.
References:
[1] Masoud, G. N.; Li, W. HIF-1α pathway: role, regulation and intervention for cancer therapy. Acta pharmaceutica Sinica. B. 2015, 5(5), 378–389.
[2] Albadari, N.; Deng, S.; Li, W. The transcriptional factors HIF-1 and HIF-2 and their novel inhibitors in cancer therapy. Expert opinion on drug discovery 2019, 14(7), 667–682.
[3] Chen, Y.; Gaber, T. Hypoxia/HIF modulates immune responses. Biomedicines. 2021, 9(3), 260.
[4] Infantino, V.; Santarsiero, A.; Convertini, P.; Todisco, S.; Iacobazzi, V. Cancer Cell Metabolism in Hypoxia: Role of HIF-1 as key regulator and therapeutic target. International journal of molecular sciences 2021, 22(11), 5703.

Round 2
Reviewer 1 Report
Manuscript was improved. However, I still have concerns as listed below.
1) Page 1, line 37: Regarding "which is a common subunit of several transcription factors", given reference is inappropriate. No description regarding ARNT subunit other than HIF-1 was provided in ref 4. Am I wrong?
2) Fig 2: I found 5) transcription and 2) DNA transcription. What is difference? Please specify. Is "transcription" inhibition of transcription factors?
3) 2-ME2 is missing in Figure 2.
4) CoCl2 and O2 should be CoCl2 and O2. Please check throughout manuscript including references.
Author Response
1) Page 1, line 37: Regarding "which is a common subunit of several transcription factors", given reference is inappropriate. No description regarding ARNT subunit other than HIF-1 was provided in ref 4. Am I wrong?
Response: ARNT is indeed not specifically described in the reference, but the description of HIF-1 mentions this passage: "The heterodimeric HIF consists of an oxygen-dependent alpha-subunit (stabilized under hypoxia) and a constitutively expressed beta-subunit (HIF-1β or ARNT). Three α subunits have been described so far: HIF-1α , HIF-2α , and HIF-3α." So, according to this passage, I summarize as "ARNT is HIF-1, HIF-2, HIF -3 common subunit". Maybe I didn't express it properly, so I revised the expression and added new references (Ref. 5 and 6). Reference 5 mentioned "All three HIFs are composed of two subunits, alpha and beta. The HIF-β subunits, also known as aryl hydrocarbon nuclear translocators (ARNT), are not regulated by any changes in oxygen” and “In contrast to HIF-β subunits, the expression levels of HIF-1α, HIF-2α, and HIF-3α subunits are tightly regulated by changes in oxygen concentration through proteolytic degradation and transcriptional regulation.” Also mentioned in Reference 6 "ARNT serves as a common binding partner for the aryl hydrocarbon receptor (AhR) as well as HIF-α subunits." Thank you very much for your comments.
2) Fig 2: I found 5) transcription and 2) DNA transcription. What is difference? Please specify. Is "transcription" inhibition of transcription factors?
Response: These are two different concepts. 5) Transcription refers to the transcriptional activity of HIF-1 downstream target genes after HIF-1α forms a transcription complex with HIF-1β, P300/CBP; 2) DNA transcription refers to the transcriptional activity of HIF-1α DNA transcribing into HIF-1α mRNA. Thank you very much for your comments.
3) 2-ME2 is missing in Figure 2.
Response: 2-ME2 has been added to Figure 2. Thank you very much.
4) CoCl2 and O2 should be CoCl2 and O2. Please check throughout manuscript including references.
Response: Modified and checked throughout our revised manuscript. Thank you very much.

Reviewer 2 Report
In the revised version authors successfully addressed most of the comments. However, in this reviewer's opinion, the authors need to add a figure of known compounds in the literature that target HIF-1a to make this review more informative for the readers.
Author Response
In the revised version authors successfully addressed most of the comments. However, in this reviewer's opinion, the authors need to add a figure of known compounds in the literature that target HIF-1a to make this review more informative for the readers.
Response: According to your suggestion, we have added Table 1 in our revised manuscript. Thank you very much for your good suggestion.
